# An Immersive Virtual Reality Training Game for Power Substations Evaluated in Terms of Usability and Engagement

**Iván F. Mondragón Bernal** [1,*,†] , **Natalia E. Lozano-Ramírez** [2,†] , **Julian M. Puerto Cortés** [1,†] , **Sergio Valdivia** [3] , **Rodrigo Muñoz** [1] , **Juan Aragón** [3] , **Rodolfo García** [4] and **Giovanni Hernández** [4]

1 Faculty of Engineering, Pontificia Universidad Javeriana, Cra. 7 No. 40-62, Bogotá 110231, Colombia; puerto.julian@javeriana.edu.co (J.M.P.C.); rmunoz@javeriana.edu.co (R.M.)
2 Department of Architecture, Pontificia Universidad Javeriana, Cra. 7 No. 40-62, Bogotá 110231, Colombia; lozano.n@javeriana.edu.co
3 Independent Researcher, Bogotá 111111, Colombia; ing.sergiovaldiviatrujillo@gmail.com (S.V.); juanklather@gmail.com (J.A.)
4 Enel-Codensa, Bogotá 110221, Colombia; rodolfo.garcia@enel.com (R.G.); giovanni.hernandez@enel.com (G.H.)
* Correspondence: imondragon@javeriana.edu.co; Tel.: +57-601-3208320
† These authors contributed equally to this work.

**Abstract:** Safety-focused training is essential for the operation and maintenance concentrated on the reliability of critical infrastructures, such as power grids. This paper introduces and evaluates a system for power substation operational training by exploring and interacting with realistic models in virtual worlds using serious games. The virtual reality (VR) simulator used building information modelling (BIM) from a 115 kV substation to develop a scenario with high technical detail suitable for professional training. This system created interactive models that could be explored using a first-person-perspective serious game in a cave automatic virtual environment (CAVE). Different operational missions could be carried out in the serious game, allowing several skills to be coached. The suitability for vocational training carried out by utility companies was evaluated in terms of usability and engagement. The evaluation used a System Usability Scale (SUS) and a Game Engagement Questionnaire (GEQ) filled by 16 power substation operators demonstrating marginally acceptable usability, with improvement opportunities and high acceptance (by utility technicians) of this system for operation training focused on safety in such hazardous tasks.

**Keywords:** electric power substation; training; immersive virtual reality; building information modelling (BIM); usability test



## 1. Introduction

Power substations are key components for the continued and reliable operation of the power system. Their outages from the power systems would severely affect the system's capability to transport power to end consumers. Normal daily operations, as well as restoring services after unplanned outages, require several manoeuvres that must be carried by qualified technicians working in hazardous environments. Several organisations focused on health and safety at work recognize electricity labour as a long-term serious workplace and public hazard, exposing employees to dangers such as electric shock, burns, fires, explosions and fatalities [1]. In the USA, the energy utility companies are in the third position in terms of accidents in the electrical area [2]. Only in Colombia, the electrical supply industry was among the most dangerous fields, with a 5/17 rank for accident rate and 2/17 for mortality rate [3]. Reported accidents' statistics in this field have been continuously growing during the last years [4]. Most reported accidents by distribution utility companies included factors such as inexperience in operation, violation of safety distances and neglect of technical standards, among others. Hence, operations training is highly important to improve safety operational performance risks and satisfy customer

needs as well as the needs of interested parties. Traditional training programs carried out by utility companies are usually based on old-fashioned teaching modes, including the study of some documentation (standards, operations and manoeuvre manuals, risk and hazardous explanations and mitigation) [5–7]. Training and qualification could include a practical experience focused on a supervised manoeuvre on real energised equipment at field (a dangerous situation for personal on training) [7]. Recently, the use of technologies such as augmented reality (AR) [8] and virtual reality (VR) [9,10] has been studied from an academic perspective to evaluate suitability in these kinds of programs. However, despite these advances, there are no published studies conducted with real substation technicians at utility companies that validated the acceptance and suitability of these new technologies to improve safety training programs.

Considering these facts, the following research question (RQ) was formulated: To what extent is an immersive virtual reality application a suitable tool to improve vocational training (in terms of usability and engagement of technicians) of power substation operations focused on safety?

## 1.1. Virtual Reality and Immersive Technologies

Virtual reality (VR) is a computer-generated scenario that simulates a realistic experience that can be interacted with in a seemingly real or physical way by a person using special electronic equipment [11,12]. It has been based mainly on interactive 3D graphics, user interfaces and visual simulation (VS) (graphical representation of objects and systems of interest using graphical languages) to display relevant data and analyses in immersive spaces. VR, in general, is extensively used in the fields of education and training due to its potential in stimulating interactivity and motivation [13]. Different studies about the use of VR and immersive systems were developed to evaluate the performance of this kind of system in a learning scenario [13,14]. For vocational training aimed at adult workers, VR offers the possibility of moving safely around dangerous places, learning to cope with emotions while experimenting with the best solutions, far away from the real dangers [13] or in those situations that could not be accessed physically, caused by limitations such as time, physical inaccessibility and ethical issues, among others.

There is a large portfolio of technologies, including interfaces, portable devices, sensors and 3D graphics, all of which are essential to achieve immersive environments in education and research [13], such as head-mounted displays (HMDs). With this system, the user experiences total immersion in the virtual world [15]. For applications where teamwork was necessary, this system does not allow the user to see the real world (only the virtual reality); therefore, the user cannot interact directly with real people, unless they interact through avatars [16]. In substation operations, tasks are usually performed in teams in response to dangerous situations, large equipment and technical guidelines of the utility maintenance crew chief. Another potential drawback is that it could cause more disorientation and visual fatigue (cyber sickness) with prolonged use compared to large projection systems [17–20]. Its biggest advantages, as compared to large screen projection systems, are a lower investment cost and the increasing availability of several equipment and tools in the market [21].

Another possible solution is an immersive virtualisation system called cave automatic virtual environment (CAVE) [12], where a physical projection space usually exists in the form of a cube or cylinder, in which various projectors with stereo capability project images to the walls and floor, surrounding several users inside the structure. The users wear active-type stereo glasses, synchronised with the projectors, allowing them to perceive stereo images within the three-dimensional scene [11,13]. In this type of space, the user can interact with people and objects inside the projection space, as the glasses do not hinder the vision and are only polarised to the images projected on the surfaces. Therefore, virtual or augmented reality applications are possible. The perception of immersion becomes possible by combining multiple projections surrounding the users and using active stereoscopy [12]. CAVE systems require a larger physical space where the projection infrastructure is installed

and, usually, their investment is greater than that of HMD systems. However, they allow several users to be present in the same projection area, whereby all can simultaneously visualise the virtual world, while maintaining the ability of interaction among each other, overcoming the disadvantages that HMD has for the specific application, requiring crew chief guidelines [16].

These technologies contribute to improving the user's immersion, presence and interactivity, as mentioned in [22]. Interactivity is the degree to which a user can modify the simulated environment. Presence is the subjective experience (illusion) of being in one place and it is closely related to immersion [23,24]. From a technological viewpoint, immersion means that the system is capable of delivering an inclusive and extensive surrounding and vivid illusion of reality. Therefore, aspects such as display resolution, stereo capability, large field of view, tracking devices and input devices contribute to this illusion [21,25]. From a psychological viewpoint, immersion is a mental and emotional state in which the user feels an isolation of the senses from the real world [22]. Therefore, immersive virtualisation technology allows the user to have the illusion of being physically, mentally and emotionally present in a virtual 3D environment [11,13].

The VR field is complemented by game-based approaches to enhance learning and training methodologies, because games are focused on entertainment and engagement [26]. Video games have demonstrated a huge potential not only for entertainment, but also to generate abilities, skills and knowledge in vocational training by exploiting the main implicit characteristics that video games already have, i.e., engagement, entertainment and influencing attitudes and behaviours, that could allow us to extend and develop physical and mental capabilities, as mentioned in the work of [27]. Using these characteristics properly could allow users to spread out the training process, as well as achieving better user concentration and involvement. This is the basis of serious games.

As mentioned in the work of [15], "*a serious game is a digital game created with the intention to entertain and to achieve at least one additional goal (e.g., learning or health)*". Serious games are complete games (or simulations) that have an educational or learning background (e.g., training). Usually, they have a mission to fulfil; therefore, this requires users to think rationally, strategise, solve problems and test different decisions during the game to obtain the best results. Serious games applications have been actively used in several areas for learning purposes, as described by [26,28], ranging from educational games for younger audiences to collaborative training and simulation environments for the industry. Some of the application fields include engineering education [29,30], health [31,32], operational skills training [28,33,34] and industrial plant operations [35,36].

However, as mentioned by [14], the mere presence of VR, immersion and educational content in a game does not guarantee its effectiveness. In fact, the educational power of any serious game also depends on various additional factors. More specifically, the educational content must be sound, age-appropriate, well integrated into the game and presented clearly and the interface must be easy to use by the target audience, increasing interactivity with the VR environment. For this reason, an appropriate methodology must be used to design a game engaging user immersion and interactivity, to speed up the process for effective learning in vocational training. Additionally, this methodology should include the generation of a realistic scenario, that is, a suitable mission according to target audience and application domain; good usability and interaction; and, finally, a tool to measure outcomes and user experience [26]. In terms of realistic scenarios, interactivity and clear presentation of educational content, the use of three-dimensional models of high precision has allowed the building information modelling (BIM) methodology to be incorporated, which involves the generation and management of data of a building or infrastructure, using dynamic software and collaborative work with a single source of the project information, as presented in Section 1.2.

*1.2. Virtual Reality and BIM Models*

BIM has been found to facilitate communication and information transmission among different stakeholders in a project [37]. BIM methodology integration with virtual reality, augmented reality and mixed reality are successful combinations to improve poor spatial cognition [38], collaboration, information processing, ability to standardise processes and reduction in risk [39–41].

BIM has been increasingly used in the infrastructure design and construction phase, because it allows high-precision, configurable, three-dimensional information models that are compatible with a variety of graphical environments, including stereoscopic (immersive) and holographic visualisation technologies, to be generated [39,42,43]. Other similar approaches are presented in [44–46]. In the study of [44], a design methodology for a virtual training system is presented. It is based on rules and typologies commonly used in a first-person point-of-view game and addresses how a BIM model is used to create a virtual environment using a game engine and creating a mission based on rules and scoring, similar to those used in video games. Similarly, the authors of [45] staged an application of computer games in design visualisation for education and a solution to address interoperability between games and building models to enhance architectural visualisation and education. The authors introduced the concept of a BIM-game prototype integrating building information modelling and gaming into architectural visualisation. Another example of a direct BIM-game system was given by [46], who integrated a BIM model and a game typology (rules) for architectural visualisation. Similarly, the work of [47] shows a process of creating a video game based on BIM modelling and interaction with a simulation of fire and smoke engines for fire evacuation training in structures, evaluating the effect of human behaviour during the evacuation process. Following a similar structure, the work of [39] presented a VR tool for BIM reviewers (VBR) for global engineering collaboration. Using an avatar, the students could immerse themselves in the BIM model and identify problems individually or collaboratively as a team. In general, these works propose different methodologies and programming flows to convert a BIM-type construction model into a model compatible with VRML (virtual reality modelling language) formats.

*1.3. Virtual Reality for Power Substations Training Programs*

Several projects have been carried out for virtual simulation of substation operations. In a work developed by [42], the implementation of a virtual environment for training in operations of an electric substation is presented. A similar method was developed in the study presented by [48], in which an expert system providing supervision and guidance in the user training process is proposed. In both cases, the simulation was only visible on a computer screen, so it did not allow users to experience total immersion perception, but it allowed the operator to know the main components of the infrastructure [49]. The work of [50] laid out a methodology for the creation of virtual worlds of electrical substations where a high adjustment to reality was proposed, using three-dimensional geometric models of high precision. With this approach, it was possible to achieve realism in the virtual world and in the training process. A similar approach was taken by [10] to generate realistic virtual environments of a power substation.

In the study presented by [44], a virtualisation system of a substation is presented, where not only a BIM model is used for the generation of the environment, but the electrical simulation of the equipment and the parameters of the electrical network are also implemented using a dynamic model programmed in MATLAB®. A software bridge was created between the graphic environment (virtual model) and the dynamic simulator in MATLAB®. The system simulated both the physical and mechanical world with the graphic engine as well as the electrical behaviour of the network, achieving great realism in training.

As previously mentioned, these studies mainly focus on the generation of realistic virtual environments or training tools at a theoretical or preliminary development. However, there have been no assessments in exploring the efficacy of these systems to motivate

technicians performing real training programs carried out by utility companies. Although there is no unified methodology to evaluate VR applications for training, as explained in the survey developed by [26], most of the works related to performance evaluation identify five key factors that must be considered: user satisfaction, learning rate, skills improvement, immersion and usability. However, this survey also indicated that studies focused on immersion and usability were scarce, although both factors had a direct correlation with a successful experience that could play a pivotal role in the learning and skill-formation process. As mentioned in [51], several studies have demonstrated the learning effectiveness of serious games. However, usability focused on game design, game flow and user interface (UI) suitable for the target audience requires additional efforts. This is a key factor because it is possible that vocational users in training could be users with no affinity to gaming. Consequently, the game interface and functionality must be accurate for a broad audience in order to maintain engagement and contribute to the training process.

To answer the RQ, the performance and suitability based on usability and engagement of an immersive serious game were evaluated as a tool to improve vocational training of power substation operations focused on safety in our study. A methodology was utilised to integrate a real power substation BIM model into a game engine, creating a realistic scenario including main power equipment information. The serious game used immersive virtualisation, seizing the advantages of a CAVE room from Pontificia Universidad Javeriana, as well as a six-degree-of-freedom (6DOF) haptic wand device, to play three operator training missions based on common tasks (maintenance, operation and failure) performed in high voltage power substations. The proposed system was validated by executing usability and engagement tests, which are suitable tools to analyse the performance of such applications. The experimental results are given for two groups of subjects, i.e., six postgraduate students of a VR course without knowledge of power systems but some experience in VR applications and 16 substation technicians with experience in power systems (with or without experience in games and VR). Finally, the problem is addressed to reduce the research gap on training process performance and quality by utility companies of technicians and operators for dangerous tasks, such as those presented in power electrical systems, aiming to lower risk, costs and grid outages, among others.

## 2. Materials and Methods

For the first stage of the training serious game, a 115/11.4 kV power substation was used as a reference for the virtual environment. This power substation was a part of the distribution ring providing coverage for the urban expansion of Bogotá D.C., Colombia. This substation was designed based on a single-busbar single-breaker (SBSB) H-scheme for the first construction phase, as shown in Figure 1. An SBSB substation consists of a single busbar with a number of incoming and outgoing lines connected to it. Each line is protected by one circuit breaker (CB). The widespread applications of this layout are distribution and transformer substations, as well as industrial feeder areas, because it is simple and cheap. The main drawback is its dependence on one main bus, causing an outage in the event of breaker or bus failure. Considering these layout factors, substation operators need to develop the ability to provide, safely and quickly, maintenance on line circuit breakers in both bay sides (high (HV) and medium (MV) voltage), in order to reduce circuit outages during normal, maintenance and failure operations.

### 2.1. Substation Building Information Modelling BIM

Based on building drawings (AutoCAD® .dwg files) and construction specifications, a BIM (Revit® 2018) model was generated for the electric substation. The model included the main structural, architectural and power equipment and control switchboards, with different levels of development (LODs), based on [52], to reduce model complexity to the specific accuracy in geometry and level of information requirements of the project, as seen in [53]. Basic construction elements were modelled using an LOD 100 (foundations, basement, site-work, etc.) and visible and main equipment and services were modelled

using an LOD 350 (interior construction, electrical services, etc.). All power equipment and control panels described previously for incoming lines, transformer bay and MV switch gears were modelled as shown in Figure 2.

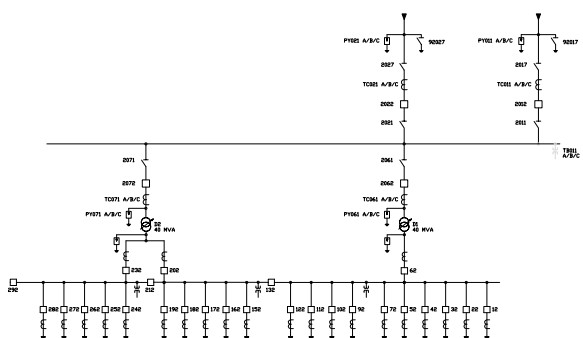

**Figure 1.** Modelled 115 kV main one-line diagram.

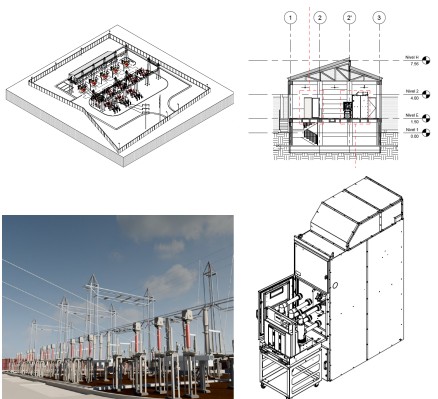

**Figure 2.** Power substation render and BIM incoming lines, transformer bay and switch gears with LOD 350.

Different methodologies have been proposed in order to use BIM in game engines [46,48,54–56]. One of these methodologies incorporates a direct integration between the game engine and the BIM design suite using of a software development kit (SDK) [54,55]. This methodology allows online modifications of BIM models to be incorporated into the game engine. However, it requires advanced SDK programming compatibility between software versions. Another methodology involves exporting a BIM file using an intermediate file adapter and then importing the appropriate model into the game engine [46]. This methodology allows BIM to be integrated fast into the game engine, but any modification of this model is not directly reflected in the game engine, forcing a new data import. For this project, the latter methodology was implemented using an open OBJ plug-in [57] for the BIM design suite. The exported file was post-processed to create a suitable FBX file format that could be directly integrated into the game engine [58], as shown in Figure 3.

### 2.2. System

The system developed for this project included several hardware systems integrated with specific software to control and play the simulator. Figure 4 shows the proposed structure. It included hardware (CAVE, MOCAP and Workstation (WS)) and software (BIM, GameEngine and middleware), explained as follows.

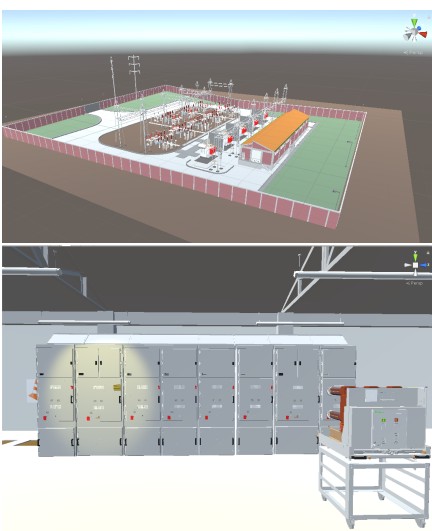

**Figure 3.** Substation rendered in Unity3D®game engine: top, full substation view; bottom, MV switchgear and spare circuit breaker in the control room.

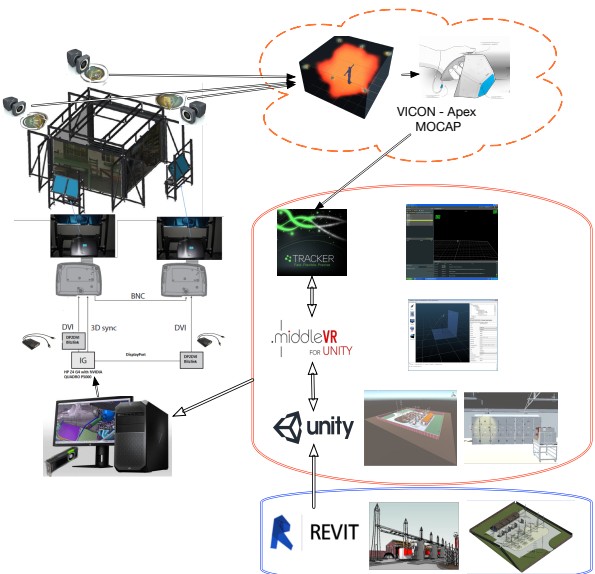

**Figure 4.** System developed for the simulator. Several hardware systems (CAVE, MOCAP and 6DOF joystick) were integrated with specific software to control and play the simulator. Such software run the game engine and integrated the imported BIM file and MOCAP data to develop missions.

### 2.2.1. Hardware

The main device used in this system was a CAVE (cave automatic virtual environment) [12,59] with immersive stereographic projection. This CAVE used high-performance DLP projectors with WUXGA resolution, active 3D and refresh rate up to 120 Hz for active stereo glasses. The CAVE was complemented with a motion capture system (MOCAP) using infrared (IR) retro reflective markers and four high-speed cameras. The MOCAP system involved a 6DOF joystick with active IR markers. This system allowed the main glasses attitude, as well as user movements and interactions, to be captured. The glasses with IR markers controlled the perspective view in the simulation, while the 6DOF joystick allowed the user virtual wand (bar interface), as well as the user 2D translation and rotation, to be controlled. All devices (hardware and software) were integrated into a single WS for fast synchronisation. This WS was an Intel Xeon Quad-Core 3.6 GHz processor with 32 GB DDR4 SDRAM, NVidia P5000 GPU with 3D stereo bracket and 512 SDD, achieving a performance of more than 45 FPS.

### 2.2.2. Software

The core of the simulation was developed using the Unity3D® (version 2018.2.1) game engine [60,61]. This engine integrated MOCAP data input through the Middle-VR® plug-in for Unity3D®. Such middleware was a library handling all aspects of a VR application, including input devices, stereoscopy, clustering, etc. Two objects were tracked using MOCAP, the main user 3D glasses and the active 6DOF haptic joystick (wand). Position and rotation w.r.t. reference coordinates in the working area for both wearable input devices were available in the game engine. The imported BIM file was used as the base simulation scenario, customising power equipment, switch gears and buildings with physical properties and interactive elements according to mission rules. Complementary ornamentation (company logos, safety signals, facilities and equipment special demarcation) was included in the base scenario, which increased the sense of realism.

### 2.2.3. Game Development

Unity 3D [61] was selected as the game engine [60], in which the entire software was developed. Based on a rapid prototyping methodology, there was a functioning prototype in a very short span, enabling the team to interact with and improve the system dynamically. Figure 5 shows the systemic approach created demonstrating the inputs of the system, the outputs and how they interacted with the MOCAP elements described previously.

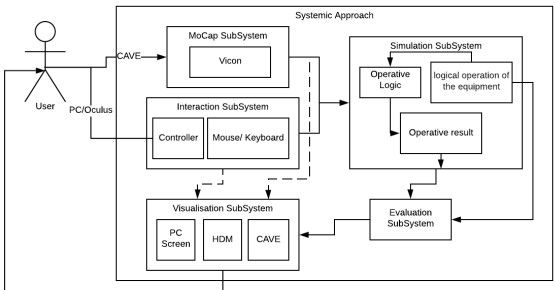

**Figure 5.** Game logic structure implemented in the game engine.

Knowing the framework that would be used and marking off the environment where the user could interact with the power substation elements, the following points were taken into account. Every object in the virtual world had to have a boundary box, where the user could not pass across the physical elements of the station, as a worker could not cross a machine in real life. To this end, the basic physics colliders provided by Unity 3D were used, giving a quick and efficient solution to create boundaries. Taking advantage of the provided colliders, interaction with the different machinery was designed so that they would be used through a ray-casting approach, where the user "pointed" to the element to interact with. This element was highlighted and, by pressing the defined button, the element performed the intended interaction. For interactions to be achievable, the system required inputs to be given at any time by the user. By definition, the simulation should work on three defined platforms, namely, a CAVE environment, an HMD and a desktop computer, thus limiting the options to the 6DOF wand controller, USB gamepad and keyboard/mouse, respectively. In this work, evaluation (see Section 3) was only performed on the CAVE version. Future works could include a comparison with the HMD version.

The game was designed following a first-person perspective with an explorative interaction typology [26]. This typology allows the user to explore and to interact freely with the virtual environment. Usually, it requires the complete development of the VR environment (which was the point of using a BIM model). Two kinds of interaction methods were defined, trying to keep simplicity and usability in an environment unknown to the test users. These methods were used to perform all the tasks in all of the missions. First, there was the "point and click" interaction, that was identified with the change in the colour

of the object. This interaction showed how the elements were displayed and how the user could understand the purpose those objects served inside the particular mission. Figure 6 (top) shows an example of an interactive whiteboard, where the mission to be performed was explained to the player (technicians must use it in real operations). The whiteboard was only filled after user interaction with this element. This was a method to guide the user during the game. Another interaction created inside the system was the "point and select" used by the operators to select an option inside a graphic user interface (GUI). These elements were displayed on screen when there was a decision to be made but no "real" element to interact with. Figure 6 (bottom) illustrates this method. In this particular case, the adequate personal protective equipment PPE had to be selected in a specific order to be able to continue with the mission. To test the performance of the users, three different missions were established within the game structure, each of one resembling real activities that sub power station operators had to perform in their daily routines. Missions had different difficulty levels, so each successful mission allowed the next mission to be performed. In the first one, the user had to perform an indoor medium-voltage switchgear maintenance on an outgoing feeder (any CBx2 in Figure 1). It included CB switching operation coordinated with the control centre and CB withdrawal and re-connection after maintenance, according to electrical utility instruction sheets and operations procedures. The second mission involved an outdoor yard equipment recognition (PY011, 2017-92017, TC011, 2011, 2012, TB011, 2061, 2062, TC061 and D1) according to the one-line diagram (see Figure 1). The final mission corresponded to an HV switchgear bay local operation of non-load dis-connector (2027-92027) coordinated with the control centre. All these missions (see Figure 7) enabled the team under training to maximise the experience by portraying all the elements present within the substation as well as safety procedures during normal, maintenance and failure operations.

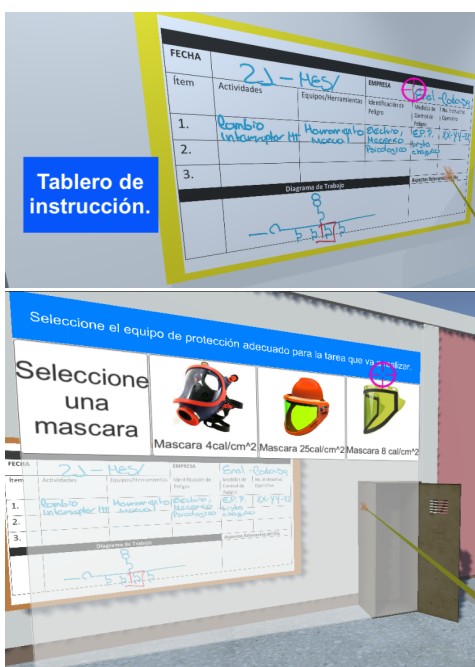

**Figure 6.** Interactions available on the game: top, the point-and-activate option can be seen where an interactive object changes its colour; bottom, the point and select are displayed.

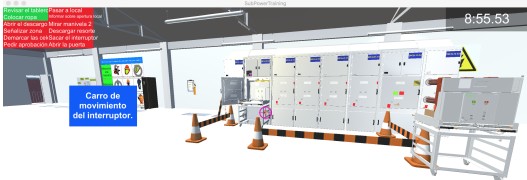

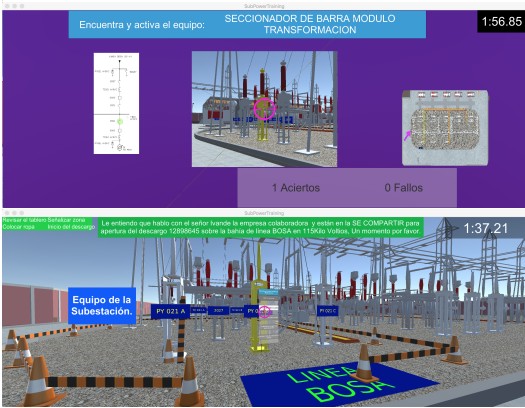

**Figure 7.** Three different missions can be performed. The top image shows indoor medium-voltage switchgear maintenance on an outgoing feeder including CB switching operation and CB withdrawal and reconnection after maintenance. The middle image shows outdoor yard equipment recognition according to one-line diagram. The bottom image shows an HV switchgear bay local operation of non-load disconnector.

In all missions, the user must complete the required task to proceed to the next assignment; at the first stage, the task to be performed was presented to the user on the screen with a one-line diagram, such as the one shown in Figure 1, and a brief mission text. Then, the user was free to explore, interact and make decisions and manoeuvres, given that users knew the standard procedures for these tasks. The user had a series of tools available, such as interaction with gear and equipment, simulated radio communication and interactions with working elements (with a visual or sound feedback), which they could freely use to carry out the proposed mission. However, players were expected to follow proper logic, regarding the security and operating procedures established by the company for the specific mission. The mission unfolded freely and continuously until the objective was fulfilled, as long as there was no risk situation or catastrophic failure ending the mission. Three basic rules were defined as the assessment tool for the user's performance:

1. The user's mission was systematically scored with the evaluation subsystem by comparing user action against defined procedures.
2. No penalties were given to the user for making mistakes as no point could be taken away from the user for making bad or wrong decisions unless it incurred into the third rule or "catastrophic error".
3. In the case of a "catastrophic error", the simulation stopped (game over), letting the user know that the error could cause life endangerment and the manoeuvre could not be completed.

The maximum score for any mission was 100, achieved by completing every step in the correct order. The score was determined by dividing the maximum score by the steps involved in every mission. Score points were determined by performing several tests for a correct selection of personal protective equipment (PPE), adequate radio communications with the control centre, operating instruction tracking and equipment manipulation, among others. The time spent executing the entire mission was also determined. After concluding the mission, users could see their scores and times compared to the top-five scores achieved by other users to promote competition, improving performance. Finally, a video of the

system can be accessed at the following link: https://youtu.be/5fKXtaPWOKA (accessed on 30 December 2021).

## 3. Evaluation

Testing and evaluation were performed using a protocol to assess both the usability of the system and the engagement in the virtual environment. These two elements are important when developing a serious game in virtual reality, as it is mentioned in [62]. In terms of engagement evaluation in the virtual environment, we used the Game Engagement Questionnaire (GEQ) [63]. The GEQ is a tool used to measure psychological engagement as a first approach to evaluate immersion within the virtual environment [63], to know if the users have shown a degree of interest and attention to the virtual process [64].

For usability purposes, the System Usability Scale (SUS) was used [65] to know how the simulator was at the time of use. The SUS is a method used to measure the usability of a product/system by asking questions to participants. Once the questions are answered (see SUS survey in Appendix A, Table A1), the responses are weighted to obtain results about the usability of the simulator. This method was chosen as it is an instrument that has been evaluated in several studies and applied in the development of projects related to new interaction environments, such as virtual reality systems, video games and simulators, among others. In the study presented by [66], the authors indicate the criteria by which the evaluation of the results was performed, assigning specific formats to the data when these were interpreted. Likewise, they explained some factors that must be considered for the subsequent interpretation and analysis of data. These factors are related to the characterisation of individuals and their experience with similar systems.

For this study, an evaluation in two phases was conducted to identify the required people to obtain a consistent result. The first group was a non-objective group to improve the system interface and acquire knowledge regarding the scope associated with its use. The sampling strategy used to develop the research was purposive sampling, where all people selected were willing to share the information and had some basic knowledge about the system and purpose of the project [67]. The second group was the focus group that validated the proposed systems. Its size was calculated from the first group using the formula to determine a confidence interval as exposed in Section 4—Results. The phases were as follows.

(1) Phase 1. Test in the non-objective group and game improvement: To evaluate the usability design within the application, a sample must be taken from the place where SUS is used. Regular students of the VR subject in a Master's program collaborated in this survey. This first phase allowed us to review the perception of the non-objective subjects (with high VR interface development knowledge) about whether the application was easy to use and manipulate. This phase attempted to define if the proposed interface allowed a clear understanding and performance of the mission to be achieved despite the accuracy of the simulation. Minor improvements on the game interface were accomplished based on this analysis. This phase was carried out as follows:

- A group of six people who did not know the process nor the mission carried out was selected to prove optimal usability.
- An explanatory introduction was provided to users, focusing on what was the work performed and what was its objective (to test the application and the mission). To achieve this, there had to be a person to guide the user within the mission and address any doubts.
- At the end of the test, users completed the SUS survey and were required to give their personal opinions in an open interview. Table 1 shows the SUS individual responses and obtained scores.
- The game interface was improved according to SUS results and user suggestions.
- The results obtained with the group contributed to the calculation of the objective group size. Based on the method used to perform the calculation, this group was a

strategic sample to achieve the standard deviation margin desired without compromising the future steps in the project.

**Table 1.** SUS individual items' response: target and non-objective groups.

| | Target Group | | | | | | | | | | | | | | | | Non-Objective Group | | | | | |
|---|---|---|---|---|---|---|---|---|---|---|---|---|---|---|---|---|---|---|---|---|---|---|
| | T1 | T2 | T3 | T4 | T5 | T6 | T7 | T8 | T9 | T10 | T11 | T12 | T13 | T14 | T15 | T16 | N1 | N2 | N3 | N4 | N5 | N6 |
| Q1 | 5 | 5 | 5 | 3 | 5 | 5 | 4 | 4 | 4 | 5 | 3 | 3 | 4 | 4 | 2 | 3 | 2 | 2 | 4 | 3 | 2 | 4 |
| Q2 | 4 | 4 | 1 | 4 | 3 | 4 | 2 | 3 | 2 | 4 | 2 | 1 | 2 | 4 | 4 | 4 | 3 | 2 | 3 | 2 | 2 | 3 |
| Q3 | 4 | 4 | 4 | 4 | 3 | 4 | 3 | 3 | 4 | 5 | 4 | 2 | 4 | 4 | 1 | 4 | 2 | 3 | 4 | 4 | 3 | 2 |
| Q4 | 3 | 3 | 4 | 2 | 3 | 2 | 3 | 4 | 3 | 4 | 4 | 5 | 4 | 5 | 5 | 1 | 5 | 4 | 5 | 4 | 5 | 4 |
| Q5 | 2 | 4 | 5 | 4 | 2 | 4 | 4 | 4 | 4 | 4 | 4 | 4 | 4 | 4 | 3 | 3 | 3 | 4 | 4 | 2 | 3 | 4 |
| Q6 | 2 | 3 | 1 | 4 | 3 | 4 | 2 | 2 | 3 | 4 | 1 | 1 | 2 | 2 | 4 | 4 | 4 | 3 | 2 | 4 | 3 | 1 |
| Q7 | 4 | 5 | 5 | 4 | 4 | 5 | 4 | 3 | 4 | 4 | 5 | 5 | 4 | 4 | 2 | 5 | 4 | 3 | 4 | 5 | 2 | 2 |
| Q8 | 3 | 4 | 2 | 3 | 3 | 5 | 2 | 2 | 2 | 1 | 1 | 5 | 2 | 4 | 5 | 3 | 4 | 5 | 2 | 1 | 3 | 4 |
| Q9 | 5 | 3 | 5 | 2 | 4 | 5 | 4 | 3 | 4 | 5 | 4 | 2 | 3 | 5 | 1 | 5 | 5 | 4 | 4 | 5 | 3 | 3 |
| Q10 | 2 | 4 | 5 | 4 | 4 | 1 | 1 | 4 | 2 | 4 | 1 | 3 | 3 | 5 | 5 | 3 | 4 | 3 | 1 | 4 | 5 | 1 |
| Score | 65 | 57.5 | 77.5 | 50 | 55 | 67.5 | 72.5 | 55 | 70 | 65 | 77.5 | 52.5 | 65 | 55 | 15 | 62.5 | 40 | 48 | 68 | 60 | 38 | 55 |

(2) Phase 2: Engagement and usability in the target group: What was evaluated was whether the simulator was attractive and kept target users engaged in carrying out mission activities. In this phase, operation technicians in the power substation evaluated the simulator. GEQ was used to determine the opinion of the test group about how the designed game was. The protocol used was similar to that of the student group but had the following differences:

- A group of 16 technicians knowing the process and familiar with the mission evaluated the system (see Figure 8). In total, 6 of the 16 were in the training process for operations in power substations but had experience in electric utility operations (promotion at work). The rest of them were trained operators with more than five years of direct experience in substations.
- An explanatory introduction to users focused on what was the work being performed and what would be its objective (to test the application and prove the mission). To perform this, there had to be a person to guide the user within the mission and address the doubts.
- At the end of the test, users completed the SUS survey and were required to give their personal opinions in an open interview. They were also asked to give a grade (from 0 to 10) to the proposed system, which was a suitable method for training power substation operators. Table 1 shows the SUS individual responses and obtained scores.
- The six technicians in training also answered the GEQ survey. In Appendix A, Table A2 shows the individual responses to GEQ questionnaire.

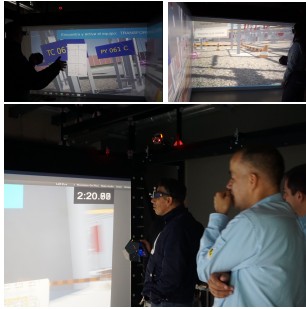

**Figure 8.** Technicians who knew the process and who were familiar with the mission that was being carried out, playing and evaluating the training serious game.

According to [68], within the interpretation of SUS scores on products, SUS scores less than 50 should be cause for significant concern and should be judged to be unacceptable;

SUS scores between 50 and 70 are marginally acceptable and should be considered as candidates for increased scrutiny and continued improvement; and SUS scores above 70 are adequate. The evaluation followed a similar methodology to that proposed by [51] and was applied to two different groups explained below.

## 4. Results

Dispersion measures were determined, obtaining an average in the weighting achieved by the non-objective group (see Tables 2 and 3) of 51.25 with a standard deviation of 11.69. According to the scale proposed by [68], the confidence of the SUS was 0.911 and the adjective "very good" could be attributed to the simulator in terms of the factor under study. The factor with the most drawbacks was the statement in the fourth question: "I think that I would need the support of a technical person to be able to use this system". This meant that, although the usability was "marginally acceptable", there were difficulties regarding the interaction with the simulator interface at first, making it difficult for new participants to have a more immersive experience within the test because these factors directly influence the user experience, thus the rating.

**Table 2.** SUS items' mean score contribution (weighted: range 0–4): non-objective group.

|      | Q1   | Q2   | Q3   | Q4   | Q5   | Q6   | Q7   | Q8   | Q9   | Q10  |
|------|------|------|------|------|------|------|------|------|------|------|
| Mean | 1.8  | 2.5  | 2.0  | 0.5  | 2.3  | 2.2  | 2.3  | 1.8  | 3.0  | 2.0  |
| S.D. | 0.98 | 0.55 | 0.89 | 0.55 | 0.82 | 1.17 | 1.21 | 1.47 | 0.89 | 1.67 |

**Table 3.** SUS scores: non-objective group.

| Mean  | S.D.  | Min | Max | Median | Usability | Learnability |
|-------|-------|-----|-----|--------|-----------|--------------|
| 51.25 | 11.69 | 38  | 68  | 51.25  | 56.25     | 31.25        |

Based on non-objective group data, it was possible to define the minimum sample size for the target group that guaranteed conclusive results. The calculation of the sample size in quantitative variables used the expression $n = \left( \frac{Z_{1-\frac{\alpha}{2}} * \sigma_x}{e_m} \right)^2$, where $Z_{1-\frac{\alpha}{2}}$ is a standardised normal distribution with a probability of $1 - \frac{\alpha}{2}$; $\alpha$, is the confidence used, given by the study conducted by [68]; $e_m$ is the mean error of the measurement, that is, the precision; finally, $\sigma_x$ is the standard deviation of the data obtained. The mean error chosen for the realisation of the tests was six units. According to this factor, the target group minimum size was 16 people. It is of vital importance to mention that, according to [62], "*a factor analysis requires a minimum of five participants per item to ensure stable factor estimates*", indicating the reliability of the case in this study despite the counterbalancing and positive or negative structure of SUS.

The mean in the weighting obtained from the target group (see Tables 4 and 5) was 60.16 with a standard deviation of 14.76, so the adjective "marginally accepted" could be attributed to the simulator according to the target group. Similar to the non-objective group, the factor with the main drawbacks was the statement in the fourth question (average of 1.56), followed by the tenth question (1.81). The question with the best evaluation was the seventh (3.19), followed by the first (3.00) and the ninth (2.75). These indicators show that the system was an useful tool for the previously targeted set; however, it required some adjustments before users felt confident with the system interface.

**Table 4.** SUS items' mean score contribution (weighted: range 0–4): target group.

|      | Q1   | Q2   | Q3   | Q4   | Q5   | Q6   | Q7   | Q8   | Q9   | Q10  |
|------|------|------|------|------|------|------|------|------|------|------|
| Mean | 3.13 | 2.13 | 2.73 | 1.67 | 2.73 | 2.47 | 3.33 | 2.2  | 2.93 | 1.93 |
| S.D. | 0.83 | 1.13 | 0.70 | 1.11 | 0.80 | 1.13 | 0.62 | 1.26 | 1.10 | 1.39 |

**Table 5.** SUS scores: target group.

| Mean | S.D. | Min | Max | Median | Usability | Learnability |
|------|------|-----|-----|--------|-----------|--------------|
| 60.16 | 14.76 | 15.00 | 77.5 | 63.75 | 64.65 | 42.98 |

## 5. Discussion

According to the results, the target group scored higher than the non-objective group, because they already had knowledge of the training area and were familiar with the real scenario in which the simulator was focused. However, it was evident that the usability of the system to improve the training process in such hazardous task was marginally acceptable, considering that the average grade (scale from 0 to 10) given by the focus group was 7.56 with a standard deviation (S.D.) of 2.45. This value could be correlated with the learnability obtained by the system. The six members in training from the target group also performed the Game Experience Questionnaire (GEQ). For the implementation of this measuring instrument, the same procedure for the SUS was performed. This instrument had many questions to guarantee a more precise approach to the individual to evaluate each of the modules making up the instrument. The GEQ was composed of four modules: core, in-game, post-game and social presence modules. For the implementation of the instrument in the simulator under study, the latter was not included because this game did not include other social entities with which the player had behavioural involvement. In Appendix A, Table A2 shows the GEQ questionnaire and the six trainers' responses. The scoring was performed following the guidelines proposed in [69] (see Appendix A, Table A3) and the results are summarised in Table 6.

**Table 6.** Game Experience Questionnaire (GEQ).

| | | | | | | | |
|---|---|---|---|---|---|---|---|
| **Core Module** | | | | | | | |
| | Operator | | | | | | |
| | 1 | 2 | 3 | 4 | 5 | 6 | Mean |
| Competence | 0.60 | 1.00 | 2.80 | 2.00 | 0.00 | 4.00 | 1.73 |
| Sensory and Imaginative Immersion | 2.33 | 3.00 | 2.83 | 2.50 | 2.33 | 1.67 | 2.44 |
| Flow | 2.40 | 2.40 | 3.00 | 2.80 | 1.40 | 2.40 | 2.40 |
| Tension–Annoyance | 0.33 | 0.33 | 0.00 | 0.67 | 3.33 | 0.00 | 0.78 |
| Challenge | 1.40 | 1.60 | 1.40 | 1.60 | 3.80 | 0.40 | 1.70 |
| Negative affect | 1.00 | 0.00 | 0.00 | 0.75 | 2.00 | 0.50 | 0.71 |
| Positive affect | 1.80 | 1.80 | 3.00 | 2.80 | 0.40 | 2.60 | 2.07 |
| **In-Game version** | | | | | | | |
| Competence | 0.50 | 1.00 | 3.00 | 2.00 | 0.00 | 3.50 | 1.67 |
| Sensory and Imaginative Immersion | 2.0 | 3.00 | 3.00 | 3.0 | 3.00 | 2.00 | 2.67 |
| Flow | 2.00 | 3.50 | 3.00 | 3.00 | 0.00 | 3.00 | 2.42 |
| Tension–Annoyance | 0.50 | 0.50 | 0.00 | 0.50 | 3.50 | 0.00 | 0.83 |
| Challenge | 2.50 | 3.50 | 2.50 | 2.00 | 3.50 | 1.00 | 2.50 |
| Negative affect | 0.00 | 0.00 | 0.00 | 1.00 | 2.50 | 0.50 | 0.67 |
| Positive affect | 1.50 | 3.00 | 3.00 | 3.00 | 0.00 | 3.00 | 2.25 |
| **Post-Game Module** | | | | | | | |
| Positive Experience | 0.83 | 1.17 | 2.50 | 2.17 | 0.33 | 1.83 | 1.47 |
| Negative Experience | 0.33 | 0.33 | 0.33 | 0.00 | 1.83 | 0.33 | 0.53 |
| Tiredness | 1.00 | 0.00 | 0.00 | 0.00 | 0.50 | 0.00 | 0.25 |
| Returning to Reality | 0.33 | 0.33 | 0.33 | 0.00 | 1.33 | 0.00 | 0.39 |

The GEQ core module assesses the users' game experience. From Table 6, it is clear that most of the operators beginning training indicated a positive affect with a good immersion and game flow feeling during the first simulation session. These results broadly agree with the in-game module data, which assessed the game experience at multiple intervals during game sessions. The post-game module, which assessed how players felt after they had stopped playing, demonstrated a positive training session with minor issues in returning

to reality. However, the GEQ instrument also showed that one of the operators did not have the best experience (during and after the test).

Based on the project development, the data and results obtained support the statement given in [62], where a slightly negative relationship between age and score was reported in the SUS. In each group, people with older age assigned lower scores in the questionnaire than the younger people.

However, several scenarios need to be taken into account to improve the system development and results. For this case, one potential improvement could be to broaden the objective population to minimise outliers in the data. Likewise, the normalisation of some data might reduce the deviation of the results, meanwhile increasing the mean. Furthermore, the increase in the number of people interviewed could add weight to the statistical significance of the conclusions suggested by [26].

Our experience has shown us that it is important to involve crew chiefs during the design process of both the simulated environment and the rules of the game. They contribute to defining which elements require greater realism, interactivity, or are important for safety-focused training. This was a factor that allowed the trainees to feel more realism. This included the use of corporate decoration and language (audio and texts) typical of the work rather than generic audiovisual material. The crew chief also contributed to motivate participants during the game experience.

It is also important to consider the technological obsolescence of this kind of VR learning environment from two perspectives—user motivation and hardware issues. As mentioned in the work of [70], training based on VR applications decreases formative effectiveness as the obsolescence process advances, because the user's motivation and immersion perception can be affected. This is a key factor; our results demonstrate that motivation is necessary to maintain involvement of the user as well as achieving the learning goal. This issue can be solved if there are continues improvements of VR applications as well as learning strategies and goals [70]. On the other hand, there is the technological obsolescence of hardware. As presented in the work of [16], there is an open discussion about the possible replacements for CAVE systems by more affordable immersive technology such as HMDs. The constant improvements in HMDs capabilities, display resolution, mobility, ease of use, costs and others make them preferable for this kind of applications, closing the gap with some of CAVEs advantages, such as social interaction and multi user experience. To avoid these issues, the system was designed to operate in these two immersive platforms, as well as desktop PCs.

Future work can focus on improving interactions between the user and the system, based on user-centred design, to increase engagement, knowledge appropriation and interconnection with daily operations. Though there were expected limitations, such as space, CAVE room availability and lack of knowledge about virtual reality from the end users, none of these prevents the created system from fulfilling the initial requirements. Moreover, it can gauge the efficacy of the operators within the designed missions, catch the attention of the test groups and provide a great baseline to create improvements on future iterations.

Regarding the research question, this virtual reality application may be considered as a suitable tool to improve vocational training, as explained in the discussion presented in this section.

VR, in general, is widely used in the fields of education and training due to its potential in stimulating interactivity and motivation. With this system, the feedback given by the operators would improve functionality further and the scores achieved for each item of both questionnaires should be used, including the likelihood question. Finally, future works could include a comparison with the HMD version.

This study explored collaborative experience by having an active and a passive player simultaneously, gaining some crucial benefits from teamwork, supervision and collaborative processes. Further research is encouraged to meet a completely collaborative experience with more than one active player concurrently.

To sum up, the project presents crucial implications for the phases implemented, despite some outliers in the data. However, it may contribute to the literature and was proven to be practical. Moreover, an extrapolation of the data recollected can reduce some errors in measurements. In the following section, general conclusions regarding the development and results of the project are presented.

## 6. Conclusions

This study focuses on analysing the performance and suitability of an immersive serious game for technician training in power substation operations aimed at safe manoeuvring. The system integrated a power substation BIM model into a game engine, creating a realistic scenario including the main power equipment. Additionally, the use of a standardised BIM model allowed us to meet the modelling requirements in both geometric and technical information.

The system was complemented with an immersive virtualisation, CAVE room and a 6DOF haptic wand device to execute three operator training missions based on common tasks performed in high-voltage power substations. The proposed system was validated using usability and engagement tests with 16 substation operations technicians. The experimental results show marginally acceptable usability, a good immersion and game flow feeling and a good subject's mean grade, demonstrating that systems of this kind are suitable for professional training of power systems but require more instances of development and improvements.

This study analyses a simulator focused on training, based on operators' safety, when performing dangerous tasks such as those presented in electrical power systems, reducing risk, costs and grid outages, among others. In the future, this system may be integrated with a real-time electric SCADA simulator, increasing the realism during power equipment switching, thus allowing the functionality of the proposed system to be expanded. Moreover, the application of BIM models might be explored even further by connecting the constructive data in model elements directly into the game by meeting full interoperability between BIM and VR software.

Additionally, it would be beneficial to compare the results obtained in this project with others in a future implementation of the same system, to evaluate additional aspects such as rate of learning and skills improvement to study the effectiveness of the test used and the correlation between them.

**Author Contributions:** Conceptualization, I.F.M.B., R.G. and G.H.; methodology, I.F.M.B., N.E.L.-R., S.V., J.M.P.C. and R.M.; software, N.E.L.-R., S.V., R.M. and J.A.; validation, S.V., J.M.P.C., R.M. and J.A.; formal analysis and investigation, I.F.M.B., N.E.L.-R., S.V. and J.M.P.C.; data curation, I.F.M.B., N.E.L.-R., S.V. and J.M.P.C.; writing—original draft preparation, I.F.M.B., N.E.L.-R., S.V. and J.M.P.C.; writing—review and editing, I.F.M.B., N.E.L.-R. and J.M.P.C.; supervision, I.F.M.B., R.G. and G.H. All authors have read and agreed to the published version of the manuscript.

**Funding:** This research study was funded by Pontificia Universidad Javeriana and the Enel-Codensa in Bogotá, Colombia, under the project "Entrenamiento virtual de manejo y operación de equipos en subestación eléctrica de potencia, mediante sistema de potencia", No. 621-2018.

**Institutional Review Board Statement:** All procedures performed in this study that involve human participants were in accordance with the ethical standards of the school of engineering committee. Participants validating this serious game were first informed about the following: (1) the purpose of the research and procedures; (2) their right to decline to participate and to withdraw from the research once participation had begun; (3) the foreseeable consequences of declining or withdrawing; (4) reasonably foreseeable factors that may be expected to influence their willingness to participate such as potential risks, discomfort, or adverse effects; (5) any prospective research benefits; (6) limits of confidentiality; and (7) whom to contact for questions about the research and research participants rights. Finally, no personal data have been recorder during this test.

**Informed Consent Statement:** Informed consent was obtained from all subjects involved in the study.

**Conflicts of Interest:** The authors declare no conflict of interest.

# Appendix A

*Appendix A.1. System Usability Scale*

The System Usability Scale used in the works is presented in Table A1.

**Table A1.** System Usability Scale survey.

| | Strongly Disagree | | | | | Strongly Agree |
|---|---|---|---|---|---|---|
| 1. I think that I would like to use this system frequently. | ☐ | | ☐ | ☐ | ☐ | ☐ |
| 2. I found the system unnecessarily complex. | ☐ | | ☐ | ☐ | ☐ | ☐ |
| 3. I thought the system was easy to use. | ☐ | | ☐ | ☐ | ☐ | ☐ |
| 4. I think that I would need the support of a technical person to be able to use this system. | ☐ | | ☐ | ☐ | ☐ | ☐ |
| 5. I found the various functions in this system were well integrated. | ☐ | | ☐ | ☐ | ☐ | ☐ |
| 6. I thought there was too much inconsistency in this system. | ☐ | | ☐ | ☐ | ☐ | ☐ |
| 7. I would imagine that most people would learn to use this system very quickly. | ☐ | | ☐ | ☐ | ☐ | ☐ |
| 8. I found the system very cumbersome to use. | ☐ | | ☐ | ☐ | ☐ | ☐ |
| 9. I felt very confident using the system. | ☐ | | ☐ | ☐ | ☐ | ☐ |
| 10. I needed to learn a lot of things before I could get going with this system. | ☐ | | ☐ | ☐ | ☐ | ☐ |

*Appendix A.2. Game Experience Questionnaire*

Table A2 shows target group responses to the GEQ questionnaire.

**Table A2.** Individual items' responses to the GEQ.

| | Core Module | | | | | |
|---|---|---|---|---|---|---|
| | Operator | | | | | |
| Statement | 1 | 2 | 3 | 4 | 5 | 6 |
|---|---|---|---|---|---|---|
| 1. I felt content | 2 | 2 | 3 | 3 | 0 | 2 |
| 2. I felt skilful | 1 | 0 | 3 | 2 | 0 | 4 |
| 3. I was interested in the game's story | 3 | 3 | 3 | 3 | 4 | 2 |
| 4. I thought it was fun | 1 | 2 | 3 | 3 | 2 | 3 |
| 5. I was fully occupied with the game | 2 | 3 | 3 | 3 | 1 | 3 |
| 6. I felt happy | 2 | 1 | 3 | 2 | 0 | 2 |
| 7. It gave me a bad mood | 0 | 0 | 0 | 1 | 4 | 0 |
| 8. I thought about other things | 3 | 0 | 0 | 0 | 3 | 0 |
| 9. I found it tiresome | 1 | 0 | 0 | 2 | 0 | 0 |
| 10. I felt competent | 1 | 1 | 3 | 2 | 0 | 4 |
| 11. I thought it was hard | 2 | 2 | 1 | 2 | 4 | 0 |
| 12. It was aesthetically pleasing | 3 | 3 | 3 | 3 | 3 | 2 |
| 13. I forgot everything around me | 3 | 3 | 3 | 3 | 0 | 2 |
| 14. I felt good | 2 | 1 | 3 | 3 | 0 | 3 |
| 15. I was good at it | 0 | 2 | 3 | 2 | 0 | 4 |
| 16. I felt bored | 0 | 0 | 0 | 0 | 1 | 2 |
| 17. I felt successful | 1 | 1 | 3 | 2 | 0 | 4 |
| 18. I felt imaginative | 3 | 2 | 3 | 3 | 3 | 2 |
| 19. I felt that I could explore thing | 1 | 3 | 3 | 1 | 3 | 2 |
| 20. I enjoyed it | 2 | 3 | 3 | 3 | 0 | 3 |
| 21. I was fast at reaching the game's targets | 0 | 1 | 2 | 2 | 0 | 4 |
| 22. I felt annoyed | 0 | 0 | 0 | 1 | 3 | 0 |
| 23. I felt pressured | 0 | 0 | 0 | 1 | 3 | 0 |
| 24. I felt irritable | 0 | 0 | 0 | 0 | 3 | 0 |
| 25. I lost track of time | 1 | 2 | 3 | 2 | 2 | 2 |
| 26. I felt challenged | 3 | 4 | 3 | 3 | 4 | 2 |
| 27. I found it impressive | 2 | 3 | 2 | 2 | 1 | 2 |
| 28. I was deeply concentrated in the game | 3 | 3 | 3 | 3 | 2 | 3 |
| 29. I felt frustrated | 1 | 1 | 0 | 1 | 4 | 0 |
| 30. It felt like a rich experience | 2 | 4 | 3 | 3 | 0 | 2 |
| 31. I lost connection with the outside world | 3 | 1 | 3 | 3 | 2 | 2 |
| 32. I felt time pressure | 0 | 0 | 1 | 1 | 4 | 0 |
| 33. I had to put a lot of effort into it | 2 | 2 | 2 | 1 | 4 | 0 |

**Table A2.** *Cont.*

| | Core Module | | | | | |
|---|---|---|---|---|---|---|
| | Operator | | | | | |
| Statement | 1 | 2 | 3 | 4 | 5 | 6 |
| In-Game version | | | | | | |
| 1. I was interested in the game's story | 3 | 3 | 4 | 3 | 4 | 2 |
| 2. I felt successful | 1 | 2 | 3 | 2 | 0 | 3 |
| 3. I felt bored | 0 | 0 | 0 | 0 | 2 | 1 |
| 4. I found it impressive | 1 | 3 | 2 | 3 | 2 | 2 |
| 5. I forgot everything around me | 3 | 4 | 3 | 3 | 0 | 4 |
| 6. I felt frustrated | 1 | 1 | 0 | 1 | 4 | 0 |
| 7. I found it tiresome | 0 | 0 | 0 | 2 | 3 | 0 |
| 8. I felt irritable | 0 | 0 | 0 | 0 | 3 | 0 |
| 9. I felt skilful | 0 | 0 | 3 | 2 | 0 | 4 |
| 10. I felt completely absorbed | 2 | 0 | 0 | 3 | 1 | 2 |
| 11. I felt content | 1 | 3 | 3 | 3 | 0 | 2 |
| 12. I felt challenged | 3 | 4 | 3 | 2 | 3 | 2 |
| 13. I had to put a lot of effort into it | 2 | 3 | 2 | 2 | 4 | 0 |
| 14. I felt good | 2 | 3 | 3 | 3 | 0 | 4 |
| Post-Game Module | | | | | | |
| 1. I felt revived | 1 | 1 | 3 | 3 | 2 | 2 |
| 2. I felt bad | 0 | 1 | 0 | 0 | 3 | 0 |
| 3. I found it hard to get back to reality | 0 | 0 | 0 | 0 | 0 | 0 |
| 4. I felt guilty | 0 | 0 | 0 | 0 | 3 | 0 |
| 5. It felt like a victory | 1 | 1 | 3 | 2 | 0 | 1 |
| 6. I found it a waste of time | 0 | 0 | 0 | 0 | 2 | 0 |
| 7. I felt energised | 1 | 2 | 2 | 2 | 0 | 2 |
| 8. I felt satisfied | 1 | 2 | 3 | 3 | 0 | 2 |
| 9. I felt disoriented | 1 | 1 | 0 | 0 | 4 | 0 |
| 10. I felt exhausted | 2 | 0 | 0 | 0 | 1 | 0 |
| 11. I felt that I could have done more useful things | 2 | 1 | 2 | 0 | 0 | 2 |
| 12. I felt powerful | 0 | 0 | 1 | 1 | 0 | 2 |
| 13. I felt weary | 0 | 0 | 0 | 1 | 2 | 0 |
| 14. I felt regret | 0 | 0 | 0 | 0 | 0 | 0 |
| 15. I felt ashamed | 0 | 0 | 0 | 0 | 3 | 0 |
| 16. I felt proud | 1 | 1 | 3 | 2 | 0 | 2 |
| 17. I had a sense that I had returned from a journey | 0 | 0 | 1 | 0 | 0 | 0 |

Table A3 shows the scoring guidelines used to obtain Table 6 from the target group responses showed in Table A2.

**Table A3.** GEQ scoring guidelines.

| Core Module | |
|---|---|
| Competence | Items 2, 10, 15, 17 and 21 |
| Sensory and Imaginative Immersion: | Items 3, 12, 18, 19, 27 and 30 |
| Flow | Items 5, 13, 25, 28 and 31 |
| Tension/Annoyance | Items 22, 24 and 29. |
| Challenge | Items 11, 23, 26, 32 and 33 |
| Negative affect | Items 7, 8, 9 and 16 |
| Positive affect | Items 1, 4, 6, 14 and 20 |
| In-Game Module | |
| Competence | Items 2 and 9 |
| Sensory and Imaginative Immersion: | Items 1 and 4 |
| Flow | Items 5 and 10 |
| Tension/Annoyance | Items 6 and 8 |
| Challenge | Items 12 and 13 |
| Negative affect | Items 3 and 7 |
| Positive affect | Items 11 and 14 |
| Post-Game Module | |
| Positive Experience | Items 1, 5, 7, 8, 12 and 16 |
| Negative Experience: | Items 2, 4, 6, 11, 14 and 15 |
| Tiredness | Items 10 and 13 |
| Returning to Reality | Items 3, 9 and 17 |

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
