# Peer review of "An Immersive Virtual Reality Training Game for Power Substations Evaluated in Terms of Usability and Engagement"

_applsci, doi:10.3390/app12020711_

Round 1
Reviewer 1 Report
Major things:
- I think the paper needs a major overhaul regarding english writing A lot of sentences do not make sense to me or are badly written. I will list a few (out of many) examples:
Lines 19-20 "Their outages from the power systems would severely affect the system’s capability to transport power to end consumers."
[better ->]
"Their failures in the power grids would affect the ability of the system to transport electricity to the end consumers."
Lines 45-80 (The research question)
"To what extent is an immersive virtual reality application a suitable tool to improve vocational training (in terms of usability and technician engagement) for substation operations with a focus on safety?"
[better->]
"To what extent is an immersive virtual reality application a suitable tool to improve vocational training (in terms of usability and technician engagement) for substation operations with a focus on safety?"
- Lines 49 - 57 are misplaced in my opinion. They should be in the end of section 1 and not in the middle of it.
- Also the order seems to be wrong…it is written about section 1, section 2.1, section 2.2 and section 5…what about section 3 and 4?
- The whole section 1.1 seems a bit odd to me. It is very far-fetched and goes back a long way to the beginnings of VR, which is rather uninteresting for the reader. Someone reading this paper should not be interested in the fact that IMUs can be used to determine the position of the HMD in virtual space. Nor that each view of the eyes is rendered with a slight offset. Theoretically, VR developers don't even need to know this because all common frameworks and engines already support this by default. This is (in my opinion) totally unrelated to the used technology which is a CAVE. To some extend it makes sense to write about presence and immersion, but then the section also writes about differences between serious games and gamification. Why? Is gamification that important to the proposed application that it is necessary to write half a page about it? I would probably remove the whole section or just keep the part about immersion/presence and CAVE specific things.
- The paper reads very tediously as it contains a lot of irrelevant information (which is not necessary for this paper). Section 1.3 is the actual relevant work while the former parts seems more like a complete thesis which tries to cover many topics but fail to properly convey the required information for the reader in my opinion.
- Section 2.2: What is meant with "CAVE, MOCAP, etc." ? This section should explain how the system works and is implemented...so what is meant with "etc."?
- The structure of the document is odd: 2.1 and 2.1.1 Describe background related topics. Section 2.2 is describing the used hardware and section 2.3 the evaluation. I would propose to use chapters instead of sections. For example, move section 2.2, 2.2.2 etc. to a chapter "implementation" and section 2.3 should be in it's own chapter.
- Chapter 3: What is the actual SUS score ? Looking at the individual numbers in table 10, I don't think it justifies the claims in the abstract "…demonstrating excellent usability…"
- Chapter 4: I have the impression that the authors try to overclaim their results. Again it is mentioned "…the usability of the system was high…" but the SUS is not mentioned here, because the SUS score is probably mediocre at best (this is the impression I get as a reader). I am not trying to judge the system here, I just think the authors should be honest in order to produce a well written paper. Even if the results are not perfect, it should just be written as is…there is no shame in saying that a system needs improvement.
- Chapter 5: The conclusion is written quite good. It feels like a different person has written it (the writing style throughout the document is not consistent). Also it is now mentioned that the system has "marginally acceptable usability" opposed to sentences before such as "…demonstrating excellent usablity" or "the usablity of the system was high".
Spelling and minor corrections:
- The abstract is written in the past "…this paper introduced…". However, when someone reads the abstract it would be better to write in present such as "..this paper presents/introduces…"
- Line 28: [?] missing reference
- Line 67: "virtual reality" is written although VR as abbrevation has been explained and is used multiple times before
- Line 76: HDMs -> HMDs
- Line 87: wa -> was
- Line 110-113: Regarding the difference between sense of presence and immersion I would recommend to add [1] and [2] as reference where the authors describe it that way: "Immersion, by its technical definition, is able to create a sensation of presence. Presence is the sense of an individual within an immersive environment and immersion stands for what the technology provides from an objective point of view"
- Line 130: Why introducing the abbrevation SG when it is not used?
- Line 177: I think sentences should not start with a reference number
- Line 231: instead of "non-gamers (do not play games often)" you could just write "users with no affinity to gaming".
- Section 2.2 Figure 3: You should not write "BIM imported in game engine" but rather clarify what engine you used
- Page 9: Is Figure 5 missing?
- Line 376: HDM -> HMD
- Line 453: You shouldn't refer to yourselves as "the authors" in the paper
- Line 484: ".." two points
Other thoughts:
- As someone from outside the field of power and electricity, sections 2.1 and 2.1.1 are also very tedious to read. I think the paper tries to fill the knowledge gap of too many readers with a drastically different background. This leaves the impression to me that the paper is not very focused. For example: As someone from the VR and HCI community, I have no real interest to read about all the specific "power hardware" that is used (e.g. IOSK-type current transformer TC061 and TC071 etc.). Someone more from the power and electricity background probably isn't interested in the specifics how a VR headset works. Maybe this paper would fit better into a "power&electricity" journal rather than "Applied Sciences".
- I don't think you should mention the creation of a "Game Design Document" at all. It is mentioned so many times with no actual benefit.
- Overall I think the document would be a much better read if it was more concise and focused without trying to "squeeze out pages" from non relevant content.
- This includes going in-depth about vr headsets although no headsets are used, the creation of a game design document, tedious explanation of the power hardware etc.
- Also the whole evaluation section spans two pages about the SUS questionnaire which could be written in a few lines and still convey all the presented information.
- Another example is line 597 - 603 where there are 6 lines written about a user having bad experiences with games in the past.
[1] Alexander Schäfer, Gerd Reis, and Didier Stricker. 2021. Investigating the Sense of Presence Between Handcrafted and Panorama Based Virtual Environments. In Mensch und Computer 2021 (MuC '21). Association for Computing Machinery, New York, NY, USA, 402–405. DOI:https://doi.org/10.1145/3473856.3474024
[2] Mestre, Daniel, et al. "Immersion et présence." Le traité de la réalité virtuelle. Paris: Ecole des Mines de Paris (2006): 309-38.
Author Response
Please see the attachment.
Thank you for your comments and suggestions. We appreciate that. Following are the answers to your comments.
|
Reviewer 1 comments |
Author's response |
|
I think the paper needs a major overhaul regarding english writing A lot of sentences do not make sense to me or are badly written. I will list a few (out of many) examples: Lines 19-20 "Their outages from the power systems would severely affect the system’s capability to transport power to end consumers." [better ->] "Their failures in the power grids would affect the ability of the system to transport electricity to the end consumers." Lines 45-80 (The research question) [better->] "To what extent is an immersive virtual reality application a suitable tool to improve vocational training (in terms of usability and technician engagement) for substation operations with a focus on safety?"
|
More of the suggestions have been incorporated in the paper, including the RQ proposal. However, we’d like to mention that the paper also has been edited and proofread by a professional service before being submitted to MDPI Journal by first time. Attached is the certificate of this service by Best Edit & Proof. Additional we also have made a full revision for proofreading for this second round. |
|
· Lines 49 - 57 are misplaced in my opinion. They should be in the end of section 1 and not in the middle of it. · Also the order seems to be wrong…it is written about section 1, section 2.1, section 2.2 and section 5…what about section 3 and 4? · The whole section 1.1 seems a bit odd to me. It is very far-fetched and goes back a long way to the beginnings of VR, which is rather uninteresting for the reader. Someone reading this paper should not be interested in the fact that IMUs can be used to determine the position of the HMD in virtual space. Nor that each view of the eyes is rendered with a slight offset. Theoretically, VR developers don't even need to know this because all common frameworks and engines already support this by default. This is (in my opinion) totally unrelated to the used technology which is a CAVE. To some extend it makes sense to write about presence and immersion, but then the section also writes about differences between serious games and gamification. Why? Is gamification that important to the proposed application that it is necessary to write half a page about it? I would probably remove the whole section or just keep the part about immersion/presence and CAVE specific things. · The paper reads very tediously as it contains a lot of irrelevant information (which is not necessary for this paper). Section 1.3 is the actual relevant work while the former parts seems more like a complete thesis which tries to cover many topics but fail to properly convey the required information for the reader in my opinion. · Section 2.2: What is meant with "CAVE, MOCAP, etc." ? This section should explain how the system works and is implemented...so what is meant with "etc."? · The structure of the document is odd: 2.1 and 2.1.1 Describe background related topics. Section 2.2 is describing the used hardware and section 2.3 the evaluation. I would propose to use chapters instead of sections. For example, move section 2.2, 2.2.2 etc. to a chapter "implementation" and section 2.3 should be in it's own chapter. · Chapter 3: What is the actual SUS score ? Looking at the individual numbers in table 10, I don't think it justifies the claims in the abstract "…demonstrating excellent usability…" · Chapter 4: I have the impression that the authors try to overclaim their results. Again it is mentioned "…the usability of the system was high…" but the SUS is not mentioned here, because the SUS score is probably mediocre at best (this is the impression I get as a reader). I am not trying to judge the system here, I just think the authors should be honest in order to produce a well written paper. Even if the results are not perfect, it should just be written as is…there is no shame in saying that a system needs improvement. · Chapter 5: The conclusion is written quite good. It feels like a different person has written it (the writing style throughout the document is not consistent). Also it is now mentioned that the system has "marginally acceptable usability" opposed to sentences before such as "…demonstrating excellent usablity" or "the usablity of the system was high".
|
We have removed lines 49-57 to reduce confusion, considering that the end of section 1 also outlines the paper structure.
Section 1.1. have been revised following your comments. Unnecessary explanation about HMD, or differences between SG and gamification have been removed.
Irrelevant information of some section has been removed.
From Section 2,2 “etc.” is removed and only implemented hardware and software are listed for the system description.
The structure of the sections is according to the guidelines for MDPI journal, that requires the following mandatory sections: Introduction, Materials and Methods, Results, Discussion and Conclusions. https://www.mdpi.com/journal/applsci/instructions
Regarding the comment for chapter 3, references have been checked to use to the same scale of qualifying. Moreover, the abstract was corrected.
In chapters 4 and 5, the adjectives used for describing the score obtained have been changed. Now each statement is congruent with the other ones. Likewise, redaction has been improved in the discussion section.
Finally, the conclusion section has been reduced according to the recommendation made by you and another reviewer. |
|
· Spelling and minor corrections: · The abstract is written in the past "…this paper introduced…". However, when someone reads the abstract it would be better to write in present such as "..this paper presents/introduces…" · Line 28: [?] missing reference · Line 67: "virtual reality" is written although VR as abbrevation has been explained and is used multiple times before · Line 76: HDMs -> HMDs · Line 87: wa -> was · Line 110-113: Regarding the difference between sense of presence and immersion I would recommend to add [1] and [2] as reference where the authors describe it that way: "Immersion, by its technical definition, is able to create a sensation of presence. Presence is the sense of an individual within an immersive environment and immersion stands for what the technology provides from an objective point of view" · Line 130: Why introducing the abbrevation SG when it is not used? · Line 177: I think sentences should not start with a reference number · Line 231: instead of "non-gamers (do not play games often)" you could just write "users with no affinity to gaming". · Section 2.2 Figure 3: You should not write "BIM imported in game engine" but rather clarify what engine you used · Page 9: Is Figure 5 missing? · Line 376: HDM -> HMD · Line 453: You shouldn't refer to yourselves as "the authors" in the paper · Line 484: ".." two points · |
Thank you for these comments. All of them have been included on the paper. Proposed references also have been added.
|
|
· Other thoughts: · As someone from outside the field of power and electricity, sections 2.1 and 2.1.1 are also very tedious to read. I think the paper tries to fill the knowledge gap of too many readers with a drastically different background. This leaves the impression to me that the paper is not very focused. For example: As someone from the VR and HCI community, I have no real interest to read about all the specific "power hardware" that is used (e.g. IOSK-type current transformer TC061 and TC071 etc.). Someone more from the power and electricity background probably isn't interested in the specifics how a VR headset works. Maybe this paper would fit better into a "power&electricity" journal rather than "Applied Sciences". · I don't think you should mention the creation of a "Game Design Document" at all. It is mentioned so many times with no actual benefit. · Overall I think the document would be a much better read if it was more concise and focused without trying to "squeeze out pages" from non relevant content. · This includes going in-depth about vr headsets although no headsets are used, the creation of a game design document, tedious explanation of the power hardware etc. · Also the whole evaluation section spans two pages about the SUS questionnaire which could be written in a few lines and still convey all the presented information. · Another example is line 597 - 603 where there are 6 lines written about a user having bad experiences with games in the past. · · · [1] Alexander Schäfer, Gerd Reis, and Didier Stricker. 2021. Investigating the Sense of Presence Between Handcrafted and Panorama Based Virtual Environments. In Mensch und Computer 2021 (MuC '21). Association for Computing Machinery, New York, NY, USA, 402–405. DOI:https://doi.org/10.1145/3473856.3474024 · [2] Mestre, Daniel, et al. "Immersion et présence." Le traité de la réalité virtuelle. Paris: Ecole des Mines de Paris (2006): 309-38. · |
In general, we have reduced some sections to improve readiness. Some of the changes included: The substation power equipment description has been reduced following your recommendation. We also have removed the mention of the Game design document, HMD detail explanation, and users' previous experiences. Also, evaluation and conclusion sections have been revised and finally some modifications requested by other reviewers to discussion section have been incorporated. We hope that these modifications improve paper quality. |

Reviewer 2 Report
The manuscript deals with a VR-based simulation tool used for safety training. It is explored, in terms of usability and engagement, in a Cave Automatic Virtual Environment (CAVE) and it incorportates Building Information Modeling (BIM). The manuscript appears in a mature state already. There are these two points which I would like to suggest for a revised version of the manuscript:
- In section 2.1, you write the following introductory sentence: “Augmented Reality and Mixed Reality were successful combinations to improve poor spatial cognition, collaboration, information processing, ability to standardise processes, reduction of risk [36–38].” No doubt, the aspect of spatial cognition is highly important to solve situations under time-pressure. The cited references, however, deal with topics of engineering and not with psychological aspects of processing virtually represented spatial information. There are ongoing debates how spatial information can be distorted and also corrected, such as distance estimations which are of high importance to solve an accurate task quickly (see for e.g. https://doi.org/10.3390/ijgi10030150 and https://doi.org/10.20870/IJVR.2009.8.1.2714)
- In your discussion section, you discuss the empirical results and refer them to (three) previous studies. As VR applications are very case-specific (your figures underline this), I wonder whether you could also recommend any design rules for an effective/efficient use of VR-environments in safety scenarios. Is the implementation of any specific visual / audiovisual feature important to increase the performance or usability? This would be another interesting contribution for the interdisciplinary community of VR designers.
Author Response
Please see the attachment.
Thank you for your comments and suggestions. We appreciate that. Following are the answers to your comments.
|
Reviewer 2 Comments |
Author's response |
|
In section 2.1, you write the following introductory sentence: “Augmented Reality and Mixed Reality were successful combinations to improve poor spatial cognition, collaboration, information processing, ability to standardise processes, reduction of risk [36–38].” No doubt, the aspect of spatial cognition is highly important to solve situations under time-pressure. The cited references, however, deal with topics of engineering and not with psychological aspects of processing virtually represented spatial information. There are ongoing debates how spatial information can be distorted and also corrected, such as distance estimations which are of high importance to solve an accurate task quickly (see for e.g. https://doi.org/10.3390/ijgi10030150 and https://doi.org/10.20870/IJVR.2009.8.1.2714)
|
This introductory section is focused on the advantages of BIM when it is combined with VR, AR on Mixed Reality. Usually, this technological combination is used in engineering applications where they have shown advantages for spatial interaction and performing a demanding task. We do not want to get deep into the psychological aspects because this paper is focused on the uses of these technologies for specific vocational training. However, we have incorporated one of the suggested references in this section.
|
|
In your discussion section, you discuss the empirical results and refer them to (three) previous studies. As VR applications are very case-specific (your figures underline this), I wonder whether you could also recommend any design rules for an effective/efficient use of VR-environments in safety scenarios. Is the implementation of any specific visual / audiovisual feature important to increase the performance or usability? This would be another interesting contribution for the interdisciplinary community of VR designers |
Following your suggestion, we have added a few lines to the discussion section with a recommended methodology to involve crew chiefs related to the field of application and realistic audiovisual material, for better acceptance of the training system. |

Reviewer 3 Report
applsci-1516981: An Immersive Virtual Reality Training Game for Power Substations evaluated in terms of usability and engagement
The authors address an interesting research topic for the journal Applied Sciences. Furthermore, it is a rigorous and well-organized paper, and the novel application is very interesting. However, some recommendations should be included in the paper:
- In my opinion, the Conclusion section should be reduced, emphasizing only the main original contribution of this research paper.
- The future works indicated in lines 643 and 659 could be included as the last paragraph in Discussion Section.
- Another idea that it would be interesting to include in the paper is the influence of technological obsolescence of this type of virtual reality learning environments. So, to avoid renumbering references a new sentence and references could be included in the Discussion Section. There is some example of paper in Applied Sciences that authors can use as a reference.
Author Response
Please see the attachment.
Thank you for your comments and suggestions. We appreciate that. Following are the answers to your comments.
|
Reviewer 3 comments |
Author's response |
|
In my opinion, the Conclusion section should be reduced, emphasizing only the main original contribution of this research paper. |
Thank you for the comment. This change has been made.
|
|
The future works indicated in lines 643 and 659 could be included as the last paragraph in Discussion Section. |
Thank you for the comment. This change has been made. |
|
Another idea that it would be interesting to include in the paper is the influence of technological obsolescence of this type of virtual reality learning environment. So, to avoid renumbering references a new sentence and references could be included in the Discussion Section. There is some example of paper in Applied Sciences that authors can use as a reference. |
A new paragraph has been included in the discussion section related to the technological obsolescence of this kind of application considering two perspectives. User motivation and hardware issues. All of them are supported by new references. Thank you for this suggestion. |

Round 2
Reviewer 1 Report
- I have no experience with Best Edit Proof, however I still think that some of the english needs improvement. If the other reviewers and editors did not notice it, it is fine for me.
In 1.0: "… is an immersive virtual reality application is a suitable tool to improve…" should be "… is an immersive virtual reality application a suitable tool to improve…"
- Table 9 accidentally moved to page 20 (within references)
- Also, I think that Appendix section should be after references? Please check if appendix comes before or after references in MDPI Applied Sciences Journal.
- Regarding the manuscript sections: I have no issue with the titles of the existing sections and chapters.
https://www.mdpi.com/journal/applsci/instructions says you need to have the required sections but not that you are not allowed to create more than stated there. I highly recommend you to at least move evaluation to its own chapter for better readability.
- Line 95: Regarding the difference between sense of presence and immersion I still recommend to add [1] and [2] as reference where the authors describe it that way: "Immersion, by its technical definition, is able to create a sensation of presence. Presence is the sense of an individual within an immersive environment and immersion stands for what the technology provides from an objective point of view"
Line 545: "… to maintain involved the user…" -> "… to maintain involvement of the user…"
Line 548: "On the other hand, is the hardware technological obsolescence" -> "On the other hand, there is the technological obsolescence of the hardware."
In general, I would recommend you to really proofread the manuscript again. For some problematic sentences you don't know if they are correct, I highly recommend you to use the translator "deepl.com" which gives very good results compared to other translation engines (I am not affiliated in any way with deepl)
[1] Alexander Schäfer, Gerd Reis, and Didier Stricker. 2021. Investigating the Sense of Presence Between Handcrafted and Panorama Based Virtual Environments. In Mensch und Computer 2021 (MuC '21). Association for Computing Machinery, New York, NY, USA, 402–405. DOI: https://doi.org/10.1145/3473856.3474024
[2] Mestre, Daniel, et al. "Immersion et présence." Le traité de la réalité virtuelle. Paris: Ecole des Mines de Paris (2006): 309-38.
Author Response
"Please see the attachment"
Thank you for your comments and suggestions. We appreciate it. Following are the answers to your comments.
|
Reviewer 1 comments |
Author's response |
|
I have no experience with Best Edit Proof, however I still think that some of the english needs improvement. If the other reviewers and editors did not notice it, it is fine for me.
|
We have not received additional comments from the other two reviewers as well as editors, regarding the English improvements. The other two reviewers have already given their final acceptance for publication. We would also like to remark that paper have been written directly in English by bilingual staff and the Best Edit Proof service, was a final step to improve readability. However, we continue to make full grammar and proofreading revisions. |
|
In 1.0: "… is an immersive virtual reality application is a suitable tool to improve…" should be "… is an immersive virtual reality application a suitable tool to improve…"
|
Thank you for this suggestion, we have applied it. |
|
Table 9 accidentally moved to page 20 (within references)
|
We apologize by this issue. This is caused by the autoformatting of the Latex package. We forced the references section to appear after table 9 and the appendix section. |
|
Also, I think that Appendix section should be after references? Please check if appendix comes before or after references in MDPI Applied Sciences Journal.
|
We have checked the authors instructions and published articles, and this is the usual structure of MDPI Applied Sciences articles to include the appendix before the references. We also have used the official Latex package, and this is the order indicated. |
|
Regarding the manuscript sections: I have no issue with the titles of the existing sections and chapters. https://www.mdpi.com/journal/applsci/instructions says you need to have the required sections but not that you are not allowed to create more than stated there. I highly recommend you to at least move evaluation to its own chapter for better readability.
|
We have moved the Evaluation to a new Section (now section 3). Thank you for this suggestion.
|
|
Line 95: Regarding the difference between sense of presence and immersion I still recommend to add [1] and [2] as reference where the authors describe it that way: "Immersion, by its technical definition, is able to create a sensation of presence. Presence is the sense of an individual within an immersive environment and immersion stands for what the technology provides from an objective point of view". [1] Alexander Schäfer, Gerd Reis, and Didier Stricker. 2021. Investigating the Sense of Presence Between Handcrafted and Panorama Based Virtual Environments. In Mensch und Computer 2021 (MuC '21). Association for Computing Machinery, New York, NY, USA, 402–405. DOI: https://doi.org/10.1145/3473856.3474024 [2] Mestre, Daniel, et al. "Immersion et présence." Le traité de la réalité virtuelle. Paris: Ecole des Mines de Paris (2006): 309-38.
|
These references have been included since revision 1 following your suggestion. Currently, they are references 23 and 24 on line 95. |
|
Line 545: "… to maintain involved the user…" -> "… to maintain involvement of the user…"
|
Thank you for this suggestion, we have applied it. |
|
Line 548: "On the other hand, is the hardware technological obsolescence" -> "On the other hand, there is the technological obsolescence of the hardware." |
Thank you for this suggestion, we have applied it. |
|
In general, I would recommend you to really proofread the manuscript again. For some problematic sentences you don't know if they are correct, I highly recommend you use the translator "deepl.com" which gives very good results compared to other translation engines (I am not affiliated in any way with deepl)
|
Thank you for sharing this translation tool with us. As mention before, the paper has been written directly in English by bilingual staff, rather than using a translation service. The Best Edit Proof service was a final step to improve readability. However, we continue to make full grammatical and proofreading revisions. |

Reviewer 3 Report
The paper was improved, and it is accepted in present form
Author Response
Thank you for your comments and suggestions. We appreciate it.
We also thank you the the final approval.